# Ethical issues in big data: A qualitative study comparing responses in the health and higher education sectors

**Annette Braunack-Mayer** [ID] *, **Lucy Carolan** [ID], **Jackie Street, Tam Ha, Belinda Fabrianesi, Stacy Carter**

University of Wollongong, Wollongong, Australia

* abmayer@uow.edu.au

## Abstract

### Introduction

The health and higher education sectors are increasingly using large administrative data-sets for secondary purposes. Both sectors experience ethical challenges in the use of big data. This study identifies and explores how these two sectors are responding to these ethical challenges.

### Objectives and approach

Through in-depth qualitative interviews, we asked 18 key Australian stakeholders using or sharing big data in the health and higher education sectors to identify the ethical, social and legal issues associated with big data use and their views on how to build ethical policies in this area.

### Results

There was strong agreement between participants in the two sectors in a number of areas. All participants believed in the benefits of data usage and recognised the importance of privacy, transparency and consent, and the duties for data custodians which followed from these principles. However, there were also significant differences. The participants in the two sectors took different views on what data are for, what benefits data should provide, who should benefit and how, and the imagined unit of analysis for working with data. Broadly, participants from the higher education sector approached these questions with individual students in mind, while health sector informants approached these questions with collectives, groups, or publics in mind. In deciding what to do, the health participants drew principally on a shared toolkit of legislative, regulatory and ethical instruments, and higher education participants on a culture of duties towards individuals.

### Conclusion / implications

The health and higher education sectors are responding to ethical challenges in the use of big data in different, but potentially complementary, ways.

**Data Availability Statement:** If the data are all contained within the manuscript and/or Supporting Information files, enter the following: All relevant

data are within the manuscript and its Supporting Information files.

**Funding:** This research is funded by a UOW 2018 SOC Seed Grant. The funders had no role in study design, data collection and analysis, decision to publish, or preparation of the manuscript.

**Competing interests:** The authors have declared that no competing interests exist.

# 1 Introduction

The term "big data" applies to the collection, linkage, analysis and sharing of very large data sets. These processes provide a powerful vehicle through which to explore elements of human behaviour, evaluate services and answer research questions. However, the sheer volume and variety of sensitive information can also lead to discrimination, loss of autonomy, infringements on privacy, reputational damage or embarrassment, identity fraud and commercial misuse of data [1].

Accounts of ethical issues in big data are proliferating across fields as diverse as educational research [2]; big health data [3–5]; big data research [6–8]; and data science in information technology [9]. These studies all highlight a similar array of ethical issues, related chiefly to privacy and confidentiality; transparency and consent; data sharing, control, access and ownership; and harms (including data misuse, vulnerabilities, group harms, discrimination and safety). Privacy and confidentiality are central concepts in this literature, with privacy generally defined as exercising control over who has information about us and under what circumstances [10] and confidentiality as the obligation not to reveal information about us outside of the context in which it is acquired [8]. The literature on big data privacy tends to assume that data will be shared, focusing on practices related to the governance of information transfers—who will have access to data and how recipients will keep data private.

In health and medicine, guidelines and codes of practice are being adapted to address the specific issues that arise in big data research [3–5]. These guidelines and codes draw on well-established and articulated codes of conduct for research practice that were initially focused on medical experimentation [11–15]. Ethical guidance for big data research in health contexts has followed in the wake of these codes: policies, practices, and infrastructure around data sharing and use have existed for some time and are being incorporated in new legislation, policy and guidance [16–20].

These new governance instruments have reach beyond health and medical research, with many countries recognising the need to regulate data collection, access, use and sharing across a wider range of settings [16–20]. The relative maturity of research ethics and big data research in health means that health research cultures have been influential in setting both cultures of research ethics broadly speaking and structures for and cultures of big data sharing and use more broadly [21, 22]. This raises questions about how well research ethics practices and policies designed initially for health and medical settings suit this wider array of settings, with scholars in the social sciences, particularly, questioning whether they are fit for purpose [23, 24].

One sector broadly comparable to health is higher education. Both feature a relatively asymmetrical power relationship between service users and service providers. In both, service users are relatively dependent on service providers for the provision of a service (care or education respectively) that can have significant impact on the life chances of the service user. In both, engagements between service providers and users lead to the creation and storage of large quantities of potentially sensitive information. For example, universities are beginning to use the data routinely collected during the student journey from application to graduation, including students' engagement with online learning systems with relatively little ethical oversight [25–27]. A recent review of student and staff perspectives on the use of big data in the tertiary education sector internationally found that, in the area of data analytics in tertiary settings, there are gaps in ethical guidance and policy, particularly with respect to inclusion of student and staff voices [28].

These historical and contemporary features suggest it may be informative to compare cultures of data ethics in health with cultures of data ethics in higher education. The purpose of

this comparison is both to provide a case study of the extent to which learnings from the health context have made their way into other contexts, and to identify potential for different fields to share innovations or enhance each other's practice. This study therefore explores how these two sectors at different stages of engagement with big data are responding to the ethical challenges attached to the use of big data. We asked key stakeholders using or sharing big data in health and higher education to identify the issues and their views on how to build ethical policy in the collection, analysis and use of big data. We aimed to better comprehend the ethical, governance and policy challenges in the use of big data in the health and higher education sectors, from the perspectives of the stakeholders. This paper compares and contrasts responses from key stakeholders in health and higher education, focusing particularly on ethical issues they identify and the practices they describe to address those issues.

## 2 Methods

### 2.1 Ethics approval

Ethics approval for the project was provided by the Human Research Ethics Committees of the University of Wollongong [2018/432] and the South Australian Department of Health and Wellbeing [HREC/19/SAH/5].

### 2.2 Approach

This study was grounded in values-based social science, or empirical ethics, using qualitative social science methods to understand the everyday moral judgements of implicated stakeholders, and using ethical theory to interpret and contextualise these empirical findings and draw normative conclusions [29, 30]. We chose key informant interviews to explore what actors in two sectors–health and higher education–thought about the ethical challenges in their work with big data.

### 2.3 Participant selection and recruitment

We purposefully recruited participants to provide insight into the ethical, social and legal issues associated with the use of big data in health and higher education. The participants were data custodians and experts in the use of large public datasets in the health sector, and information technology (IT), researchers and teaching specialists in learning analytics and/or data analytics in the higher education sector in Australia. We identified participants using three methods: recommendations from members of our expert reference group; evidence from published work indicating that the participant had done original work on the subject; and public profile or organisational role that evidenced expertise (eg data custodian). In health, stakeholders were recruited through their involvement with publicly funded organisations involved in data linkage of large government databases with a specific focus on health data, including the Population Health Research Network (PHRN), [16]. In higher education, potential participants were identified through research we had previously conducted on staff and student attitudes to the use of big data in this sector [28]. Potential participants were identified from those with contact details that appeared in the public domain. Potential participants in both sectors were contacted by email with an information sheet and consent form.

All the participants from the health sector and most of the participants from the education sector indicated that they had considered at some depth the issues associated with the protection and use of personal data collected in the systems they managed. Most described extensive discussions with colleagues about these issues. In the health sector this reflected widespread debate, across professional networks, about the ethical issues; in the education sector, most

participants suggested that such in-depth consideration and debate was rare and confined, at the time of interview, to a small number of tertiary education institutions who employed recognised change champions in the area.

All participants were familiar with the use of very large administrative datasets with thousands to millions of records. Health sector participants drew on their experience with research using health data routinely collected in hospitals and allied health services for patient care. They also referenced linkage between these health datasets and routinely collected administrative data held in education, justice and welfare government departments. Higher education participants focused on demographic, financial and academic data collected about students during their studies, and also referred to the increasing use of behavioural data from online learning management systems to track student engagement and performance. Regardless of the sector, for the most part data subjects in Australia rarely provide consent for the information in these datasets to be used for secondary purposes such as research, although they are likely to be informed that their information will be used for quality assurance purposes. There is evidence, from the health sector, that the Australian public is aware that medical records may be used for research purposes [31] but it is unclear how aware students are of secondary uses of their university administrative data.

## 2.4 Data collection

One author (JMS) conducted the semi-structured interviews. Using open-ended questions, we asked participants to reflect on big data practice and policy in their workplace and previous relevant workplaces, identify examples of best practice and problematic practice in their areas and institutions, and provide recommendations for change (see interview schedule in online material). We sought their perspectives on how the use of large data sets has evolved over time in their institution and how governance structures and institutional policy had changed to address challenges along the way. We asked questions about current ethical issues in the use of big data, and sought examples of justifiable and unjustifiable ways of using big data for both research and service provision. The interviews were fully recorded and transcribed.

## 2.5 Data analysis

We managed the data in N-Vivo and employed an iterative and reflective analytical process. LC openly coded the interviews, with JMS coding two of the interviews in parallel to ensure coding integrity. LC and ABM then developed a coding framework that incorporated the initial codes under higher level concepts with a particular focus on the ethical issues participants described.

As we undertook more analysis, similarities and differences between the views of participants in the two sectors became obvious, particularly with respect to the ethical issues they identified and the practices they described to address those issues. Our final coding framework focused explicitly on these similarities and differences.

All participants consented to the general use of their interview responses in the study. However, given the sensitivity of the topic, some of our participants were apprehensive about the possibility that they and their organisations might be identifiable. To address these concerns, we assured participants that they would be contacted to obtain their explicit consent for the use of quotations if their quotations were used in the paper. Prior to publication, we tried to contact all participants whose quotations appear in this paper. We were able to reach 15 of 17 participants and received approval from all to include the selected quotations. No quotations from participants we could not reach have been included in the paper, but their data have been

included in the analysis. We have also reported the participant characteristics across both sectors to provide an additional level of anonymity.

## 3 Findings

Between February 2019 and March 2020, we conducted 17 semi-structured interviews each of approximately 1 hour in length, with 18 key stakeholders, nine in each of the health and higher education sectors. One of the interviews in the health sector was conducted with two participants. All of the participants, in their working role, had previously reflected on the social and ethical issues associated with the use of big data, and readily volunteered responses, which were extensive and nuanced. Table 1 describes the participant characteristics.

### 3.1 Ethical issues in the use of big data

We found that, broadly speaking, the range of ethical issues that participants in both sectors reported were similar. However, how participants described these issues and why they were issues varied, aligned to the sector in which they worked. Sometimes, these issues were clearly labelled as ethical problems, using the language of traditional research ethics—for example, privacy, confidentiality, consent, harms and benefits. At other times, the participants described their concerns in technical or practical terms—for example, identifiability, who controls data access, or security–appreciating that there were ethical implications. Below we lay out the key issues participants described and the similarities and differences between the sectors. Table 2 summarises the ethical issues participants raised, how those issues were being addressed and proposals to address them in the future.

   **3.1.1 Privacy and confidentiality.**   The participants in this study did not draw a clear ethical distinction between privacy and confidentiality, using the terms interchangeably and without definition. They described these issues in terms of practices, focusing particularly on problems with de-identification and data sharing and access.

**Table 1. Participant characteristics.**

| Participants | N = 18 | | |
|---|---|---|---|
| Gender | | Background and Employment history* | |
| Male | 12 | *more than one per participant* | |
| Female | 6 | | |
| | | Information Technology | 4 |
| Age (years) | | Data Analytics | 2 |
| <39 | 2 | Learning Analytics | 6 |
| 40–49 | 6 | Education Technology | 3 |
| 50–59 | 8 | Tertiary & Education Research | 5 |
| 60–69 | 2 | Sciences | 1 |
| State | | Data Custodian, Governance, & Linkage | 7 |
| New South Wales | 6 | Medical Research & Clinical Trials | 2 |
| Victoria | 0 | Ethics | 4 |
| South Australia | 4 | Social Sciences | 3 |
| Western Australia | 2 | Economics, Maths & Statistics | 2 |
| Queensland | 0 | | |
| Tasmania | 1 | | |
| ACT | 4 | | |
| Northern Territory | 1 | | |

**Table 2. Ethical issues and practices to address them as described by participants.**

| Ethical Concerns | Sector | Raised by participants | What is being done | What should be done |
|---|---|---|---|---|
| Privacy and confidentiality 1. Identifiability 2. Data ownership, access & sharing | Health | 1. yes 2. yes | • National legislation & regulation • Governance frameworks • Agreed procedures & formal data management systems (Five Safes Framework, separation principles, data minimisation) | • Refine existing legislation, governance & procedures • Legislation & penalties for data misuse |
| | Higher Education | 1. yes 2. yes | • Local policies & procedures to limit data access | • Formal data management systems & procedures • National legislation, regulation & governance frameworks • Education & training for staff and students |
| Transparency | Health | yes | • Collaboration with government • Community engagement (surveys, public forums, websites) | • More public communication & education re value of data & uses |
| | Higher Education | yes | • Student facing dash boards | • More communication with students • Repeated consent • Public information • Student data literacy |
| Informed consent | Health | no | Not discussed by participants | Not discussed by participants |
| | Higher Education | yes | • Consent strategies | • More frequent consenting • Transparency re data use and users |
| Surveillance/ stigmatisation | Health | yes | • Agreed procedures & formal data management systems (Five Safes Framework, separation principles, data minimisation) | Refine existing legislation, governance & procedures • Legislation & penalties for data misuse |
| | Higher Education | yes | • Limit data access • Data minimisation | • Train algorithms • Limiting data access • Data minimisation. |
| Loss of trust in institutions | Health | yes | • National legislation & regulation• Governance frameworks• Agreed procedures & formal data management systems (Five Safes Framework, separation principles, data minimisation) • Community engagement | • Development of social licence • More public communication |
| | Higher Education | no | Not discussed by participants | Not discussed by participants |
| Realising benefit from increased data usage | Health | Yes: Public benefit | • National legislation & regulation • Governance frameworks • Agreed procedures & formal data management systems (Five Safes Framework, separation principles, data minimisation) | • Increase data sharing across borders • Share information • Increase training |
| | Higher Education | Yes: Student benefit | • Academic staff access to data • At risk students identified & contacted | • Share information • Increase training • Better strategies to identify at risk students |

*Identifiability.* Participants in both sectors believed that data in the fields in which they worked can never be permanently de-identified and, therefore, anonymity cannot be guaranteed, particularly in very large datasets containing many data points about each individual. Some participants framed de-identification of individuals as a myth; others noted that identification of groups could have negative consequences.

*Sometimes we can't guarantee privacy but where we're not prepared to guarantee privacy, then we should be at least transparent. (U7)*

*I mean, the computing capacity has changed immensely, so that's obviously been a huge thing. But the other thing we've learnt that we weren't sufficiently cognisant of at the beginning and*

*what was really in retrospect obvious at the beginning, or should've been, is that the more things you link the greater chance, greater possibility there is to reidentify people. (H8)*

There was general agreement from both sectors that more should be done to protect privacy. *Data ownership, access and sharing.* Who received data, and how, were concerns in both sectors. Common concerns were that some people overstepped their clinical or service roles to use data for other purposes without permission, or that data may be shared with untrustworthy entities, including those outside of Australian Government funding or Australian legal jurisdiction:

*It is about the ideas that clinicians have, and I better say most clinicians who are researchers, that because they have access to patient information as clinicians, they don't see a difference between accessing that data for clinical purposes as opposed to research purposes. (H1)*

*There have been a lot of new players who have moved into the higher education sector world-wide who are capturing information about students, potential students, and given that higher ed is one of Australia's biggest exports. . .the protection of that personal information that those often international organisations hold is of interest to me, because we have thousands of students who are registered with training and education institutions who are not Australian government funded or guided. (U2)*

There were also observable differences between health and education informants. First, health sector participants believed that people should their own data, but expressed concern that data custodians believed themselves to be data owners; higher education participants were less clear about data ownership:

*And that's what I think governments forget—not everybody, of course. It's our data, not the government's data, but there's an attitude that says once an organisation has it, it's their data and they therefore don't . . . owe any responsibilities back to the original source, shall we say? (H9)*

*So I would say that the data governance right now is a loosely distributed affair, but owners of particular big areas are still the owners of that data so can get access to a student's private information. (U4)*

Second, health sector participants generally had more concerns about access to datasets: for example, that data were shared with people who were not competent to analyse them, or that data were being stored in unsafe ways:

*I don't think this particular team of researchers can really analyse this much data sensibly . . . there tends to be a few. . . probably more in the community services space than the health space, and there has been the occasional project that gets funded, and approved, because there's nothing sinister about it but you do think, "Someone's already done this" but there doesn't seem to be a system that can really check on that. (H3)*

*Of course, some people are moving to actually holding all of that data in the cloud so again there's obviously technical and financial reasons for doing that but I think that is a huge security risk and, I'm personally very uncomfortable about moving to the cloud. (H8)*

**3.1.2 Transparency and consent.** Participants from both sectors endorsed a need for greater transparency about data uses, including risks and benefits.

*Transparency is important. I think it's particularly important at the point at which you collect the data. I'm thinking now of cohort studies. I'm thinking of surveys, but also administrative data collections. I think people need to have that trust at the point they give data by and large. It's interesting, people do generally trust researchers to be acting in the public interest. (H7)*

However, the participants in the two sectors differed in the way they approached the problem of transparency, including with whom data users should be transparent and, consequently, how this related to consent. Broadly speaking, health participants assumed that individual consent for big data use was unnecessary; it was impossible to seek consent from individuals and the permission for data use rested with suitable authorities. What was necessary, and often lacking, was not consent but good public information about data use and adequate legislative oversight.

*And there is one that is I think is still kind of an open sore which is that there is all this . . . data that is collected without anyone ever giving any consent for it and no really critical appraisal of whether that is something that is reasonable to waive. It is collected under conditions of power really; it is the responsible minister in each jurisdiction that gives the consent on behalf of the people. (H4)*

*But I think often the public benefit isn't communicated well enough to the public and I think that was the case again, probably, arguably with the My Health Record about. . . those potential benefits as opposed to the risks . . . there are risks, you can't deny them and you can't avoid them. But I would certainly like to see as much emphasis placed on the outcomes that may arise from linkage of their data. (H5)*

Higher education participants, in contrast, assumed that individual consent was important: failures of transparency were failures to achieve individual consent, and students needed to know how their information was being used so they could make decisions about what information they shared. They reported that students did not have enough information, the consent was too broad, or it was not obtained at all. They wanted students to consent to the use of their data but they could also see that individual consent for every use of student data was not practicable. Some participants addressed the dilemma by noting the broad consent provided on enrolment and emphasising the need for better student awareness at this point and throughout their student journey.

*I mean the whole issue with privacy and data is a tricky subject and you can have countless debates about it, I can put one hat on and say no we need to be completely transparent. They need to provide consent before they enrol in every single course. At the same time, I can put another hat on and say wait a minute, does this actually make sense to do? (U6)*

*When students enrol, they sign a student declaration and in that it specifically says that they authorise the university to. . .collect and release their personal and sensitive information in accordance with the confidentiality of students' personal information and university privacy policy. (U6)*

*And certainly, there are situations where we can't guarantee consent but again, you know, in a sense, we need to let people know that by enrolling in the university, you are consenting to something and therefore we won't be seeking your consent every time we do this from now on. So just knowing how your data is going to be used, I think, should be a fundamental principle. (U7)*

**3.1.3 Harms.**    Participants from both sectors spoke about the harms that could arise from the use of big data. Both groups noted that linking datasets could lead to increased surveillance and stigmatisation. Health sector participants also emphasised how loss of public trust could undermine research and health care.

*Surveillance and stigmatisation.* In both sectors, there were participants apprehensive that linking big data sets was enabling a "surveillance state". Higher education informants were especially concerned about tracking student movements and behaviours in order to push services, communications or marketing to them. In both education and health, concerns about surveillance were heightened for already-vulnerable or stigmatised groups (including First Nations peoples and students with mental health problems or from non-English speaking backgrounds or specific neighbourhoods).

*They are the worries of colonisation, who wants to know whether I am Indigenous or not, who wants to know where I live, how many people live in my house, who wants to know what they are going to do. . . so that is pretty common across all Australia, and New Zealand and Canada and Argentina. That is easy enough to deal with. It is just mistrust of authority that . . . if it is possible to identify a community that is almost like identifying a person. (H4)*

*But one of the things that would be recorded in the CRM [Customer Relationship Management] would be if a student had a mental health issue for example and has sought help from university services in that area. And, we know that some people have biases around that which could lead to judgemental attitudes and so on. So not having that information available to academics is something that we have protections around at the moment. But it's another one of those complex issues around complete access to data versus not. (U7)*

*Loss of trust in institutions.* Participants from the health sector identified loss of trust in government, the health system and private companies as a harm, recognising that public trust was both essential and fragile. While their principal emphasis was on the impact that loss of trust can have on data collection, and therefore on research quality, some participants also noted that individuals might not seek services if they lost trust in how the service provider managed their data.

*Well, what are you going to do, what are you going to do when people opt out of getting health care, because, you know, a lot of people don't want this to happen and if they don't trust the government to work with the data that they're entrusted with, then they just won't give you the data? (H1)*

**3.1.4 Realising benefits from data usage.**    Finally, participants in both sectors said that their organisations needed to make more and better use of data if the benefits were to be realised. The health sector participants focused on public benefits, explaining that big data research played an important, if sometimes indirect, role in improving health outcomes, and noting here the benefits of public health surveillance. The higher education participants mainly wanted to see big data used to help individual students at risk of failing or in need of support. Both groups were concerned that not sharing data would mean that these benefits could not be achieved. They identified two barriers to sharing: the lack of enabling legislation and policy, and a fear of negative publicity amongst data custodians.

*You talk about smoking and heart disease and behavioural risk factors. And they're real genuine successes, but to sort of look at the research you're doing today and point at it and say,*

*"This will be the expected benefit to the public." Well, it's a body of evidence, it will probably be of benefit, but it's not a direct A to B to C benefit. And it would be good if we could spend more time collectively, policymakers, researchers, everyone in this chain, showing what benefit research is. (H7)*

*I think there's real traction, real potential but it has to be aligned to learning–and within that–and I think that that's almost an ethical piece in itself. How are we constructing what learning is and understanding it and I think we've got a duty to use the data. . . (U9)*

*. . .it's not secret stuff and it shouldn't be secret because . . . it's using public money; it should be very transparent . . .If we say something that could be misinterpreted out of context and then a newspaper runs an alarmist story. . . . . .people become more reluctant to share data and this actually inhibits all the good we're trying to do with it. (H2)*

## 3.2 What to do about ethical issues in big data

We asked the participants to describe how they were addressing the ethical issues they had identified and what they thought could or should be done in the future. Their responses fell into two main groups of strategies–procedural strategies and communication strategies.

**3.2.1 Procedural strategies.** Participants in both sectors described procedural strategies for addressing ethical issues in big data. However, there were stark differences between the groups with respect to how legislation, policy and procedure informed their views. The health sector participants all referenced the national and state legislation and ethical guidance that regulate the collection, sharing and use of datasets in Australia. They also described governance frameworks and practical techniques that were used across the sector to reduce data flows to a minimum and to limit who could have access to identifying information about people. A number of participants mentioned the Five Safes Framework [32], a tool to assess risk associated with data sharing and release, and practices such as the data separation principle [29] which requires that the people who have access to identifying information about an individual (to undertake the linkage) do not have access to any other information about that individual.

*. . .you need to rely on the Five Safes system which is going to be about safe data, safe settings and safe outputs. And then the safe people and the safe projects, which we're lucky in that we've got the Ethics Committee to provide guidance about what is a safe project and what is a safe person. (H6)*

*The point is to be able to justify for every single record, to sort of reduce the number of records, to reduce the number of variables that you've been given. To even, sometimes it comes back to the levels of aggregation. Do you need age? You know, a date of birth. Well, ethics committees will be reticent to hand over a date of birth. The next step on the continuum might be age in years. The next step might be age groups in five years. And, each time you take a step, the ethics committee is more likely to approve it because it is less of a risk to individual privacy. But, of course, you need to refer back then to your research question. (H7)*

When the health sector participants looked to the future, they thought that, by and large, these frameworks would continue to serve them well, although they could do with some adjustment or refinement. Where there was potential for change, it would be to address particularly challenging areas, such as cross-jurisdictional data flows, complex data linkage and release of data to the private sector. They suggested sharing information about data linkage and more training for researchers as strategies to improve data flows.

*So I think it would be useful if there was a section maybe within the Commonwealth Privacy Act about linked data explicitly. That might be a way of dealing with it. But I think there does need to be some legislation. . .for misuse. (H8)*

By contrast, participants in the higher education sector repeatedly spoke about the *lack* of principles, systems and policies to guide staff working with big data. They were concerned that the sector seemed to lack leadership in this regard, relying instead on sensible individuals to make good decisions in a vacuum.

*We don't have any code of practice or any guidelines yet and I think that, in fact, that's something that is sitting. . . there was no university policy or statement on how to use the data. And so, now I can still have these conversations on linked data which is exciting and freeing but you're relying on good judgement [of individuals]. . . (U9)*

*We need some ethical frameworks, so maybe it doesn't quite fit into research, maybe it fits into operations. But there is no way that I think that I should just be able to do that as a lone ranger; and I wouldn't be. I would get into trouble for that. But where do I go, and who do I go [to] with this idea and who is overseeing it and what principles are they using? To me, that is actually a pretty simple operational fix really. But it requires awareness, expertise, strategy, a staff plan. (U8)*

Filling the ethical guidance vacuum was a problem. Some education participants considered that greater local institutional oversight and guidance were needed, whereas others suggested that it would be helpful to have national policies, frameworks, and governance, perhaps informed by policy and practice in the health sector. Overall, there was a sense of incremental policy shift as universities identified challenges and responded to them in an ad hoc manner.

*So it needs to be performed and owned at all levels, at that very individual, academic, and that school, but I think the governance piece and then the determination of boundaries of use– I think that needs to be high level. (U9)*

*I do think that [is] what we can learn from the way that data is managed in the health sector. And having some kind of really easy, big picture principles that you could use to help guide you, I think that's attractive. I don't know what they'd be but I think something like that would be incredible. . .I often think that, medical and health ethics is often a little bit more advanced than learning and teaching, university kind of ethics. I think the thing is with medical research, is that it's quite common that there is a negative consequence to a participant or a risk to a participant and it's about weighing the risk against the benefit. (U7)*

**3.2.2 Communication strategies.** Participants in both sectors sought to address ethical challenges by proposing more and better ways to interact with the people who were providing data. However, they spoke quite differently about how to do this. As noted above, health sector participants began from the assumption that contact with individual data owners simply was not possible. In its place, they wanted to establish a stronger social licence, through more effective communication with whole communities, to explain how people's data were protected and to build public trust in the organisations that held those data.

*We can't hold conversations with individuals within a big data set but you can try and engage with community. (H1)*

*And the other option we've looked at is running, and it's still on our radar, is to actually run public forums throughout the state introducing not only what we do but data linkage more broadly and with people understanding, having that chance to be able to, I guess, provide feedback much more broadly on what they understand data linkage to be and to raise any ethical concerns or other concerns about the use of their data. (H5)*

*I think, regardless, we have to be really conscious of demonstrating competence when it comes to the use of particularly administrative data, linked data. (H7)*

Where the health participants assumed they could not communicate directly with individual data owners, higher education participants assumed that they could and should. Building in part on their preference for individual consent, they described a range of ways in which individual students could learn more about how their data were used, including periodic re-consent and regular feedback to students through existing communication channels.

*So, for example, some of the student-facing dashboards out there, they collect data on how many times students go to the library and they show that on the dashboard to the students–you've attended the library this many times. (U6)*

*So, you know, the notion of periodically re-engaging with them to remind them of what they've consented to or require them to re-consent on an annual basis, that sounds good but again, you know, if it turns out that it's a 20 page document and you just sign it at the bottom because right now I want to do something and I can't do it until I click 'I agree', then it doesn't achieve anything really. (U7)*

## 4 Discussion

To our knowledge this is the first attempt to make a cross-sectional comparison of the culture of data use in two sectors in the same jurisdiction. Health and higher education are relevant sectors to compare as they are both data-intensive: our analysis reveals some important commonalities, but also critical differences. We will consider these commonalities and differences, then propose lessons that could be learned between the two sectors.

Strong commonalities existed in practical understandings (arising from common expertise) and in normative commitments. Most fundamentally, all participants believed in the benefits of data usage. All recognised the importance of privacy, while understanding that true anonymity is impossible. All valued, in principle, transparency and consent, and ascribed duties to data custodians, including to share data only with permission, and to share only with trustworthy recipients. A commonly held concern was the potential for data sharing and use to enable a surveillance state, with a high chance of discrimination. These commonalities suggest some common ground between those who work with data for a living across sectors but the differences between the health and higher education sectors are also instructive.

Participants from the two sectors took very different views on what data are for, what kinds of benefit data should provide, who should benefit and how, and the imagined unit of analysis for working with data. Broadly, participants from the higher education sector approached these questions with individual (identifiable) students in mind, while health sector informants approached these questions with collectives, groups, or publics in mind.

To explain: in higher education, data subjects were imagined as individual people, and the unit of analysis was the individual student; informants often located themselves as university staff with personal obligations towards these identifiable university students. Data were a resource, to be used to benefit students by providing them with services as customers, and also

by identifying them when they were vulnerable to provide necessary academic support and pastoral care. Thus higher education informants combined commercial and contract-based reasoning (competition through better service and contracts between service providers and purchasers) with more ethically-inflected duties, and these duties were towards identifiable individuals with whom the speaker had relationships (contractual, community or pastoral). One of these duties was to identify and support students who were underperforming, at risk of failing or in some kind of need. Another was to gain valid consent from individual students to use their data, and to communicate with individual students about data use. The final duty related to minimising the potential for harm, which was conceptualised as potential harm to individual students via, for example, privacy violations, or stigmatisation, bias or unjust discrimination.

Health sector informants took a very different position. They did not connect with data subjects as individual people to whom they might have personal obligations. Data subjects were more abstracted entities, and the unit of analysis was the public, or at least, groups within or sectors of the public. Some health sector informants were critical of clinicians who used their service delivery roles to access data precisely because it blurred the line between data subjects and individual patients. Although the original purpose of data collection may have been patient care, their goal as data custodians was to use big data to provide public benefit broadly conceived–not to provide any particular benefit to any identifiable individual. The potential harms of concern inhered at a collective level, such as loss of trust in institutions. Finally, approaches to consent and transparency focused strongly on publics or communities. Informants simply assumed that gaining consent from, or even communicating with, individuals was impossible, instead advocating for better public communication, and gaining legitimacy through endorsement from appropriate decision-makers.

These contrasting underlying logics—using data to service, support and benefit individuals versus using data as a public resource to benefit publics—connected to the different data cultures and practices in the two sectors, including different shared commitments regarding what it means to hold, use and share data in an ethically justified way. Broadly speaking, health sector informants took a research-oriented approach, and were confident in a range of well-understood and authoritative procedures to address potential ethical problems. This was consistent with a decade or more of public frameworks and infrastructure for data use, including centralisation of oversight, clear definition of common frameworks and cooperation for common goals. (One exception, as noted above, was that some clinicians failed to understand that the data generated in their own clinical practice belonged not to them, but to their patients, and this was presented as a problem that needed addressing). In contrast, universities were creating their own approaches to data use, and higher education sector informants were often concerned about a lack of common procedures, which left disparate and not always senior individuals to make decisions on their own. Where an institution had a common approach to data use, it was generally because a champion within that institution had led the development of the approach.

We note that both sectors are in fact governed by the same legislative, regulatory and ethical instruments, [33–38] with health data generally have a higher level of privacy protection in legislation. Although these instruments govern both health and educational data, one sector was acutely aware of this governance, while the other appeared much less aware. The health sector informants placed ethical data governance at the centre of their organisations' activities and expected their colleagues to do likewise. By contrast, many of the higher education informants implied that they would struggle to find others in their institutions who shared their interest in ethically informed policy for the management of big data. This indicates the importance of

more proximal procedures, policies and institutions in giving these instruments practical application and importance.

In practice, data ownership is contentious, with some advocating that data subjects are data owners [39] and others arguing that the legal concept of ownership is unhelpful, and the focus should instead be on concepts such as control or [8] looser models for connecting people with their data [40]. While this remains unresolved, for our purposes here what is of most interest is a somewhat counterintuitive difference that existed between the two sectors on the issue of ownership. Higher education informants clearly imagined individual students as the unit of analysis, but did not regard them to be data owners: they were either unclear about data ownership, or assumed universities owned data. Health informants thought at the level of publics, but believed that individual patients or research participants were data owners, and that custodians held data on their behalf (and they sometimes expressed frustration with those in the health sector who did not understand or respect this principle). This mismatch underscores that data cultures are central but not always internally coherent, or necessarily consistent with what the law requires. The culture of duties towards individuals, including duties of care, was strong in higher education: the health sector could benefit from more often remembering that those providing data are often patients requiring care. Although this study focuses on ethical issues in data intensive research, the ethical issues raised by participants in this study are not necessarily unique to the management of big datasets. The potential for identification, for example, is always a concern in research studies, regardless of the size of the dataset. By contrast, transparency is important for all research, but it is a significantly greater challenge for big data research: the size of the participant pool and the fact that data are often collected with a waiver of consent make it difficult to be confident that participants understand that their information is being used for secondary purposes.

A strength of our study was that we recruited expert informants with a nuanced understanding of how big data were being used in their sector in Australia. These informants spoke candidly and in detail about their experiences. Our comparison of the similarities and differences between the higher education and health sectors also highlights in a new way characteristics that may be taken for granted within each sector. A limitation is that, as with all qualitative research, our participants were not necessarily representative of the broader group of stakeholders from which they were drawn. However, the issues that our participants identified within each sector are consistent with findings from other studies [3, 9, 28], which supports the rigor and trustworthiness of our findings [41].

## 5 Conclusion

Recent legislative changes to regulate data collection, access, use and sharing highlight both the volume of activity occurring in this area and the level of public interest [16–20]. Aligning institutional responses to regulatory and public expectations will therefore be increasingly important [42–45]. However, as our study shows, even within a single regulatory environment, institutional cultures play an important role in defining what counts as an ethical issue in big data and what can and ought to be done about it. Our study shows the importance of being attentive to both centralised standards-setting and local institutional culture in establishing ethical practice in the use of big data.

## Acknowledgments

We would like to acknowledge the assistance of the PHRN in recruitment; A/Prof Xiaoqi Feng, UNSW School of Population Health for assistance in study design; and the participants for their candid and informative contributions.

## Author Contributions

**Formal analysis:** Annette Braunack-Mayer, Lucy Carolan.

**Funding acquisition:** Annette Braunack-Mayer, Jackie Street.

**Investigation:** Belinda Fabrianesi.

**Methodology:** Belinda Fabrianesi.

**Writing – original draft:** Annette Braunack-Mayer, Lucy Carolan, Jackie Street, Stacy Carter.

**Writing – review & editing:** Annette Braunack-Mayer, Lucy Carolan, Jackie Street, Tam Ha, Belinda Fabrianesi, Stacy Carter.

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
