## [Decision Letter · Decision Letter 0]

17 Oct 2022

PONE-D-21-33047Ethical issues in big data: A qualitative study comparing responses in the health and higher education sectorsPLOS ONE

Dear Dr. Carolan,

Thank you for submitting your manuscript to PLOS ONE. After careful consideration, we feel that it has merit but does not fully meet PLOS ONE’s publication criteria as it currently stands. Therefore, we invite you to submit a revised version of the manuscript that addresses the points raised during the review process.

Two reviewers have evaluated your submission, and have identified a number of significant concerns that need to be carefully addressed. Please pay particular attention to providing the methodological clarifications the reviewers have requested.

We look forward to receiving your revised manuscript.

Kind regards,

Jamie Males

Editorial Office

PLOS ONE

Journal Requirements:

1, Please ensure that your manuscript meets PLOS ONE's style requirements, including those for file naming. The PLOS ONE style templates can be found at

2. Please clarify the consent procedures used and measures to ensure anonymity. You mention that you contacted participants before publication for consent to publish their data, please clarify whether you obtained consent from all participants to publish their data and/or quotes.

Reviewers' comments:

Reviewer's Responses to Questions

**Comments to the Author**

1. Is the manuscript technically sound, and do the data support the conclusions?

Reviewer #1: Partly

Reviewer #2: Yes

2. Has the statistical analysis been performed appropriately and rigorously? 

Reviewer #1: N/A

Reviewer #2: I Don't Know

3. Have the authors made all data underlying the findings in their manuscript fully available?

Reviewer #1: Yes

Reviewer #2: Yes

4. Is the manuscript presented in an intelligible fashion and written in standard English?

Reviewer #1: Yes

Reviewer #2: Yes

5. Review Comments to the Author

Reviewer #1: Thanks for this opportunity to review this manuscript. Some suggestions are provided in the followings for polishing this manuscript:

1. In Introduction, examining ethical issues surrounding big data should be focused on those who potentially or actually using big data for some purposes. Some previous academic work has been reported to support focusing on higher education sector. However, the importance of focusing on health sector and the comparison between health sector and higher education sector are not reported. As such, the theoretical framework for bolstering this qualitative in-depth interview study is weak. Please revise and report.

2. For conducting a qualitative in-depth interview study, the key for sampling is to invite information-rich participants to be interviewed. In this study, the participants invited were stakeholders such as experts, data custodians in health sector, and information technology (IT), researchers and teaching specialists in learning analytics and/or data analytics in the higher education sector. However, no sufficient information can convince the readers that these stakeholders were information-rich people. Please provide more information, if there is any, to let the readers know that all these participants provided rich information during in-depth interview.

3. This is confusing. The authors reported that their participants implicitly or explicitly referenced this way of thinking about research ethics in their interviews, and also reported Principles of Biomedical Ethics. This is no doubt that, for research ethics, the three principles in Belmont Report, i.e. respect for person, beneficence, and justice, are not exactly the same as the four principles in biomedical ethics, i.e. autonomy, beneficence, maleficence, and justice. The four principles of biomedical ethics are usually followed when examining the ethical issues in medical care, between patients and physicians. In comparison, three principles of research ethics proposed in Belmont Report are usually followed in conducting research. Why did the authors refer to four principles of biomedical ethics as the framework when analyzing the data collected for ethical issues surrounding big data? Please report.

4. Was there any work done for triangulating this qualitative study results? If yes, please report how triangulation was done for convincing the future readers that this study results are scientifically convincing and sound.

Reviewer #2: This is an interesting and worthwhile survey. It is, however, vague or, perhaps better, not discerning as regards certain key concepts and issues. Chief among these:

1. It would be well to know more about the data sets used by each cohort, including how big they actually are. It might be that the survey and issues it raises would be similar if small data sets were at issue. What, that is, about big data that changes the various foci and concerns? Put differently, it is assumed or implied that the move from small or middling data sets to big data is a difference in kind – but this should be addressed.

2. In Table 2, the fact that the health cohort did not note consent as an issue is peculiar – or might not be if we knew more about why the data were collected in the first place. But if we knew that, it might illuminate the role of consent. Were the data collected during clinical encounters? Hospital monitoring? Biomedical research? We are told “it was impossible to seek consent from individuals …” (p. 10, line 259), but why is that the case? Many health care organizations seek such consent, and others disclose that de-identified data will be saved and analysed. As for relations between participants and the sources of data in their sets, it is noted they “assumed they could not communicate directly with individual data owners” (p. 15, lines 459f). This, too, is going to be dependent on the circumstances of data collection. In many studies, for instance, patients are told whether they will be re-contacted or not; in others, such re-contact would be quite difficult. Generally, it is a curiosity why there is this discrepancy between the health and education sectors. We should know more about the kinds of data sets at issue in both cases

3. It might also be that the data were collected for public health purposes as part of legitimate epidemiologic surveillance… yet surveillance is cast here as an unacceptable wrong. Indeed, there is a literature suggesting that Australians know that legitimate authorities collect public health data, and support such collection.

4. If it is true that “both sectors are in fact governed by the same legislative, regulatory and ethical instruments” (p. 17, lines 534f), those instruments should be identified. It could be noted that in many jurisdictions around the world, different laws govern educational and health data, and there is often tricky overlap.

Certain assumptions are stipulated, and likely should be softened:

P. 8, line 186: “data can never be permanently de-identified” – there is indeed a large amount of research on privacy-protecting software and some of it makes re-identification quite difficult if not impossible. And does impossibility alone suffice to justify data sharing?

p. 9, line 219: “people owned their own data…” – this is a large issue and some would argue that “control” is a far better method of managing data than the legal concept of ownership. In fact, some have argued that no one owns patient data and information, including patients.

Comments on the text:

P2, line 51 reads “The term ‘big data’ describes….” Indeed, terms do not describe. Label? Identify? Applies to?

P 13, lines 376ff: It would help if “Five Safes” and the “data protection principle” were better identified, explained or described. Note that the transcripts would do with closer editing. There is for instance no good reason to capitalize “Five Safes” in the text but not in the transcript.

6. PLOS authors have the option to publish the peer review history of their article (what does this mean?). If published, this will include your full peer review and any attached files.

Reviewer #1: No

Reviewer #2: No

---

## [Author Response · Author response to Decision Letter 0]

19 Jan 2023

January 2023

Dear PLOS One Editorial Office

Re Response to Review of Manuscript

PONE-D-21-33047

Manuscript Title: Ethical issues in big data: A qualitative study comparing responses in the health and higher education sectors

Thank you for reviewing our manuscript and for providing the opportunity for us to respond. Please find below our response to your requests for additional information and /or modifications. Please note, the PLOS One queries/ comments are in bold and our responses are below each one. 

Thank you, we have adhered to Plos One’s guidelines

2. Please clarify the consent procedures used and measures to ensure anonymity. You mention that you contacted participants before publication for consent to publish their data, please clarify whether you obtained consent from all participants to publish their data and/or quotes.

All participants whose quotes are included in the paper have given permission for their quotations to be used. We have clarified this in the text by adding material to the relevant paragraph:

All participants consented to the general use of their interview responses in the study. However, given the sensitivity of the topic, some of our participants were apprehensive about the possibility that they and their organisations might be identifiable. To address these concerns, we assured participants that they would be contacted to obtain their explicit consent for the use of quotations if their quotations were used in the paper. Prior to publication, we tried to contact all participants whose quotations appear in this paper. We were able to reach 15 of 17 participants and received approval from all to include the selected quotations. No quotations from participants we could not reach have been included in the paper, but their data have been included in the analysis. We have also reported the participant characteristics across both sectors to provide an additional level of anonymity.

Thank you, we have adhered to PLOS ONE’s guidelines and included ORDID numbers for authors

Reviewers' comments:

5. Review Comments to the Author

Reviewer #1: 

1. In Introduction, examining ethical issues surrounding big data should be focused on those who potentially or actually using big data for some purposes. Some previous academic work has been reported to support focusing on higher education sector. However, the importance of focusing on health sector and the comparison between health sector and higher education sector are not reported. As such, the theoretical framework for bolstering this qualitative in-depth interview study is weak. Please revise and report.

We have rewritten most of the introduction (section 1) to make the framework and rationale for this study clearer. Briefly, we are suggesting that governance structures for data use in health have been extensively examined, debated, developed and instituted in formal guidelines. These governance frameworks developed for health and medicine have influenced other sectors, with concerns about their applicability outside of the health setting. Data use in higher education provides an excellent point of comparison because both sectors create and store large amounts of data, both have a relatively asymmetrical power relationship between service users and service providers, and both provide services which can have a significant impact on people’s life chances. Our aim in this study was to compare the different cultures of data ethics in the two sectors. 

2. For conducting a qualitative in-depth interview study, the key for sampling is to invite information-rich participants to be interviewed. In this study, the participants invited were stakeholders such as experts, data custodians in health sector, and information technology (IT), researchers and teaching specialists in learning analytics and/or data analytics in the higher education sector. However, no sufficient information can convince the readers that these stakeholders were information-rich people. Please provide more information, if there is any, to let the readers know that all these participants provided rich information during in-depth interview.

Thank you for picking this up. We have addressed this request in two ways with additional text added to section 2.3 ‘Participant selection and recruitment’. First, we have provided an overarching statement about the methods we used to identify participants:

We identified participants using three methods: recommendations from members of our expert reference group; evidence from published work indicating that the participant had done original work on the subject; and public profile or organisational role that evidenced expertise (eg data custodian). 

Second, we have added additional comments about the participants’ depth of understanding of these issues to indicate that these stakeholders were information rich participants:

All the participants from the health sector and most of the participants from the education sector indicated that they had considered at some depth the issues associated with the protection and use of personal data collected in the systems they managed. Most described extensive discussions with colleagues about these issues. In the health sector this reflected widespread debate, across professional networks, about the ethical issues; in the education sector, most participants suggested that such in-depth consideration and debate was rare and confined, at the time of interview, to a small number of tertiary education institutions who employed recognised change champions in the area.

3. This is confusing. The authors reported that their participants implicitly or explicitly referenced this way of thinking about research ethics in their interviews, and also reported Principles of Biomedical Ethics. This is no doubt that, for research ethics, the three principles in Belmont Report, i.e. respect for person, beneficence, and justice, are not exactly the same as the four principles in biomedical ethics, i.e. autonomy, beneficence, maleficence, and justice. The four principles of biomedical ethics are usually followed when examining the ethical issues in medical care, between patients and physicians. In comparison, three principles of research ethics proposed in Belmont Report are usually followed in conducting research. Why did the authors refer to four principles of biomedical ethics as the framework when analyzing the data collected for ethical issues surrounding big data? Please report.

Thank you for picking this up and we agree that the reference to Beauchamp and Childress is not all that helpful because the results are not really structured around the four principles; accordingly, we have removed it. Our analysis was more strongly structured around the participants’ talk and we have revised the data analysis section (2.5) to make this clear. 

4. Was there any work done for triangulating this qualitative study results? If yes, please report how triangulation was done for convincing the future readers that this study results are scientifically convincing and sound.

While we share the reviewer’s concern about the validity and reliability of research, we also note that there is considerable disagreement in the qualitative research community on the value and practice of triangulation. We sought to assure the quality of our research through:

• Having multiple coders: two researchers (JS and LC) coded two interviews in parallel initially (Investigator triangulation);

• Having multiple data analysts: two coders (LC and ABM) refined the coding framework through repeated reconsideration of the text, working both independently and together (investigator triangulation); and

• Comparing our findings to existing literature, including the scoping review we had previously undertaken on ethical issues in the secondary use of data in the higher education sector (data source triangulation). 

Reviewer #2: 

This is an interesting and worthwhile survey. It is, however, vague or, perhaps better, not discerning as regards certain key concepts and issues. Chief among these:

1. It would be well to know more about the data sets used by each cohort, including how big they actually are. It might be that the survey and issues it raises would be similar if small data sets were at issue. What, that is, about big data that changes the various foci and concerns? Put differently, it is assumed or implied that the move from small or middling data sets to big data is a difference in kind – but this should be addressed.

All participants in this study were familiar with very large administrative datasets, with thousands to millions of records. While some may also have had experience with ‘small to middling’ datasets, the focus in the interview was on large to very large datasets. We have included the following sentence in 2.3 ‘Participant selection and recruitment’ to clarify this:

All participants were familiar with very large administrative datasets with thousands to millions of records.

In addition, we appreciate the prompt to consider whether the issues our respondents identified, and the solutions, were specific to big data, or whether they might apply also to ‘small to middling data sets’. We have added some comments in the discussion (section 4) to address this issue:

Although this study focuses on ethical issues in data intensive research, the ethical issues raised by participants in this study are not necessarily unique to the management of big datasets. The potential for identification, for example, is always a concern in research studies, regardless of the size of the dataset. By contrast, transparency is important for all research, but it is a significantly greater challenge for big data research: the size of the participant pool and the fact that data are often collected with a waiver of consent make it difficult to be confident that participants understand that their information is being used for secondary purposes. 

2. In Table 2, the fact that the health cohort did not note consent as an issue is peculiar – or might not be if we knew more about why the data were collected in the first place. But if we knew that, it might illuminate the role of consent. Were the data collected during clinical encounters? Hospital monitoring? Biomedical research? We are told “it was impossible to seek consent from individuals …” (p. 10, line 259), but why is that the case? Many health care organizations seek such consent, and others disclose that de-identified data will be saved and analysed. As for relations between participants and the sources of data in their sets, it is noted they “assumed they could not communicate directly with individual data owners” (p. 15, lines 459f). This, too, is going to be dependent on the circumstances of data collection. In many studies, for instance, patients are told whether they will be re-contacted or not; in others, such re-contact would be quite difficult. Generally, it is a curiosity why there is this discrepancy between the health and education sectors. We should know more about the kinds of data sets at issue in both cases.

Thank you for this comment. Taken together with comment 2 above, we have realised that we should have provided more context about the way in which the data our participants were describing are collected and used. We have provided an additional paragraph in the recruitment section to clarify this:

All participants were familiar with the use of very large administrative datasets with thousands to millions of records. Health sector participants drew on their experience with research using health data routinely collected in hospitals and allied health services for patient care. They also referenced linkage between these health datasets and routinely collected administrative data held in education, justice and welfare government departments. Higher education participants focused on demographic, financial and academic data collected about students during their studies, and also referred to the increasing use of behavioural data from online learning management systems to track student engagement and performance. Regardless of the sector, for the most part data subjects in Australia rarely provide consent for the information in these datasets to be used for secondary purposes such as research, although they are likely to be informed that their information will be used for quality assurance purposes. There is evidence, from the health sector, that the Australian public is aware that medical records may be used for research purposes (Research Australia 2020) but it is unclear how aware students are of secondary uses of their university administrative data.

The discussion has an explanation for why there was less interest in consent amongst health sector participants. 

3. It might also be that the data were collected for public health purposes as part of legitimate epidemiologic surveillance… yet surveillance is cast here as an unacceptable wrong. Indeed, there is a literature suggesting that Australians know that legitimate authorities collect public health data, and support such collection.

Thank you for picking this up. We have added a comment in the section on realising the benefits of data linkage to note that the health sector participants did recognise the benefits of public health surveillance:

The health sector participants focused on public benefits, explaining that big data research played an important, if sometimes indirect, role in improving health outcomes, and noting here the benefits of public health surveillance.

However, participants also talked about the potential for harms to arise, particularly for vulnerable populations and we have included this in the discussion. 

4. If it is true that “both sectors are in fact governed by the same legislative, regulatory and ethical instruments” (p. 17, lines 534f), those instruments should be identified. It could be noted that in many jurisdictions around the world, different laws govern educational and health data, and there is often tricky overlap.

We have included references with the quotation above (section 5) to the key legislative, ethical and regulatory instruments in Australia and noted that, although similar laws govern educational and health data, health data generally have a higher level of privacy protection. 

Certain assumptions are stipulated, and likely should be softened:

P. 8, line 186: “data can never be permanently de-identified” – there is indeed a large amount of research on privacy-protecting software and some of it makes re-identification quite difficult if not impossible. And does impossibility alone suffice to justify data sharing?

This comment is the participants’ views about the contexts in which they were working. We have amended the relevant sentence to make this clearer:

Participants in both sectors believed that data in the fields in which they worked can never be permanently de-identified…

We note in the discussion that, despite their concerns, all participants believed in the benefits of data sharing.

p. 9, line 219: “people owned their own data…” – this is a large issue and some would argue that “control” is a far better method of managing data than the legal concept of ownership. In fact, some have argued that no one owns patient data and information, including patients.

Thank you for this. This comment is the participants’ views about their data. We have amended the manuscript at this point to read:

First, health sector participants believed that people should own their own data…

We have also mentioned in the discussion (section 5) that this is a vexed issue:

In practice, data ownership is contentious, with some advocating that data sources are data owners (Riso et al 2017) and others arguing that the legal concept of ownership is unhelpful, and the focus should instead be on concepts such as control (Richards & King, 2014) or looser models for connecting patients with their data (Ballantyne 2020). While this remains unresolved, for our purposes here what is of most interest is a somewhat counterintuitive difference that existed between the two sectors on the issue of ownership…

Comments on the text:

P2, line 51 reads “The term ‘big data’ describes….” Indeed, terms do not describe. Label? Identify? Applies to?

Thank you – we have amended this.

P 13, lines 376ff: It would help if “Five Safes” and the “data protection principle” were better identified, explained or described. Note that the transcripts would do with closer editing. There is for instance no good reason to capitalize “Five Safes” in the text but not in the transcript.

Thank you for noting the typographical error – we have corrected this. We have briefly explained the Five Safes Framework and the data separation principle in the manuscript where they appear. Both terms are referenced to lengthier explanations. 

A number of participants mentioned the Five Safes Framework (AIHW, 2020) a tool to assess risk associated with data sharing and release, and practices such as the data separation principle (Davies & Dunn, 2015) which requires that the people who have access to identifying information about an individual (to undertake the linkage) do not have access to any other information about that individual.

We thank you again for your time and look forward to your feedback. 

Best wishes Lucy Carolan

For Prof Annette Braunack-Mayer

University of Wollongong

References

Australian Institute of Health and Welfare (AIHW). The Five Safes Framework: Australian Government, 2020 [Available from: https://www.aihw.gov.au/about-our-data/data-governance/the-five-safes-framework.

Ballantyne A. How should we think about clinical data ownership? J Med Ethics. 2020;46(5):289-94.

Davies R IJ, Dunn M. . A systematic review of empirical bioethics methodologies. BMC Medical Ethics 2015;2015;16(15.)

Research Australia, Public opinion poll on health and medical research and innovation. 2020.

Richards NM, King J. Big Data Ethics. Wake Forest Law Review. 2014;49(2):393-432.

Riso B TA, Vears DF, Felzmann H, Cockbain J, Loi M, Kongsholm NCH, Zullo S, Rakic V,. Ethical sharing of health data in online platforms - which values should be considered? Life Science Social Policy. 2017;Aug 21;(13(1)).

---

## [Decision Letter · Decision Letter 1]

14 Feb 2023

Ethical issues in big data: A qualitative study comparing responses in the health and higher education sectors

PONE-D-21-33047R1

Dear Dr. Carolan,

We’re pleased to inform you that your manuscript has been judged scientifically suitable for publication and will be formally accepted for publication once it meets all outstanding technical requirements.

Kind regards,

Kyaw Lwin Show, MPH

Academic Editor

PLOS ONE

Additional Editor Comments (optional):

Reviewers' comments:

Reviewer's Responses to Questions

**Comments to the Author**

1. If the authors have adequately addressed your comments raised in a previous round of review and you feel that this manuscript is now acceptable for publication, you may indicate that here to bypass the “Comments to the Author” section, enter your conflict of interest statement in the “Confidential to Editor” section, and submit your "Accept" recommendation.

Reviewer #1: All comments have been addressed

Reviewer #2: All comments have been addressed

2. Is the manuscript technically sound, and do the data support the conclusions?

Reviewer #1: Yes

Reviewer #2: Yes

3. Has the statistical analysis been performed appropriately and rigorously? 

Reviewer #1: N/A

Reviewer #2: I Don't Know

4. Have the authors made all data underlying the findings in their manuscript fully available?

Reviewer #1: Yes

Reviewer #2: Yes

5. Is the manuscript presented in an intelligible fashion and written in standard English?

Reviewer #1: Yes

Reviewer #2: Yes

6. Review Comments to the Author

Reviewer #1: The reviewer's comments have been all addressed. The revision according to the reviewer's comments is scientifically sound and convincing.

Reviewer #2: The authors have done a good job in addressing comments.

7. PLOS authors have the option to publish the peer review history of their article (what does this mean?). If published, this will include your full peer review and any attached files.

Reviewer #1: **Yes: **Yen-Yuan Chen

Reviewer #2: No

---

## [Editor Report · Acceptance letter]

21 Feb 2023

PONE-D-21-33047R1 

Ethical issues in big data: A qualitative study comparing responses in the health and higher education sectors 

Dear Dr. Carolan:

I'm pleased to inform you that your manuscript has been deemed suitable for publication in PLOS ONE. Congratulations! Your manuscript is now with our production department. 

Kind regards, 

on behalf of

Dr. Kyaw Lwin Show 

Academic Editor

PLOS ONE